# AAV Immunotoxicity: Implications in Anti-HBV Gene Therapy

**DOI:** 10.3390/microorganisms11122985

**Published:** 2023-12-14

**Authors:** Ridhwaanah Jacobs, Makafui Dennis Dogbey, Njabulo Mnyandu, Keila Neves, Stefan Barth, Patrick Arbuthnot, Mohube Betty Maepa

**Affiliations:** 1Wits/SAMRC Antiviral Gene Therapy Research Unit, Infectious Diseases and Oncology Research Institute (IDORI), Faculty of Health Sciences, University of the Witwatersrand, Parktown 2193, South Africa; 2Medical Biotechnology and Immunotherapy Research Unit, Institute of Infectious Disease and Molecular Medicine, Faculty of Health Sciences, University of Cape Town, Cape Town 7700, South Africa; dgbden001@myuct.ac.za (M.D.D.);; 3South African Research Chair in Cancer Biotechnology, Department of Integrative Biomedical Sciences, Faculty of Health Sciences, University of Cape Town, Cape Town 7700, South Africa

**Keywords:** adeno-associated viral vectors, gene therapy, hepatitis B virus, immunotolerance, immunotoxicity

## Abstract

Hepatitis B virus (HBV) has afflicted humankind for decades and there is still no treatment that can clear the infection. The development of recombinant adeno-associated virus (rAAV)-based gene therapy for HBV infection has become important in recent years and research has made exciting leaps. Initial studies, mainly using mouse models, showed that rAAVs are non-toxic and induce minimal immune responses. However, several later studies demonstrated rAAV toxicity, which is inextricably associated with immunogenicity. This is a major setback for the progression of rAAV-based therapies toward clinical application. Research aimed at understanding the mechanisms behind rAAV immunity and toxicity has contributed significantly to the inception of approaches to overcoming these challenges. The target tissue, the features of the vector, and the vector dose are some of the determinants of AAV toxicity, with the latter being associated with the most severe adverse events. This review discusses our current understanding of rAAV immunogenicity, toxicity, and approaches to overcoming these hurdles. How this information and current knowledge about HBV biology and immunity can be harnessed in the efforts to design safe and effective anti-HBV rAAVs is discussed.

## 1. Introduction

Viral infections caused by “old” viruses, such as hepatitis B virus (HBV), and epidemics or pandemics caused by “new” viruses, such as severe acute respiratory syndrome coronavirus 2 (SARS-CoV-2), also known as COVID-19, continue to threaten human existence. It is estimated that 296 million individuals are chronically infected with HBV, the etiological agent of hepatitis B. Chronic hepatitis B (CHB) results in about one million deaths annually owing to HBV-related liver complications such as cirrhosis and hepatocellular carcinoma (HCC) [1]. 

Vaccination remains the primary mechanism to reduce HBV-related mortality. The first available vaccines targeting HBV were plasma-derived and were produced by harvesting the subvirion particles of the HBV surface antigen (HBsAg) from the plasma of chronic HBV asymptomatic donors. In 1986, second-generation HBV vaccines, consisting of yeast-derived recombinant HBsAg, replaced plasma-derived vaccines. Currently, recombinant DNA vaccines, composed of highly purified HBsAg adsorbed to an aluminum adjuvant, are being used for universal HBV immunization [2,3]. After three doses of the HBV vaccine, a long-term immune response can be achieved in most immune-competent individuals. However, 5–10% of those who are vaccinated are non-responders. A protective response is defined as anti-HBsAg antibody levels of ≥100 IU/L; however, in the case of vaccine ‘non-responders’, their anti-HBsAg antibody levels are lower than 10 IU/L [4,5,6]. While the molecular mechanisms responsible for non-responsiveness remain largely unclear, some may include the failure of antigen presentation and insufficient production of T helper subset 1 (Th1) and 2 (Th2) cytokines upon vaccination. Several known factors contributing to non-responsiveness include certain genetic risk factors, a compromised immune response, chronic diseases such as diabetes, and being older than 30 years [7,8]. Efforts to develop novel vaccines include the use of RNA- and viral vector-based approaches. RNA lipid nanoparticle (LNP) formulations and adenovirus (AdV)-based vaccines have performed well in pre-clinical studies [9,10,11]. Recombinant adeno-associated viral vectors (rAAVs) have also been explored for vaccine development. However, as a result of the inherent low immune stimulation, rAAVs are not favored for vaccine development [12,13]. Nevertheless, they are promising candidates for antiviral immunotherapy [14,15]. 

The fact that anti-HBV drugs, comprising nucleoside/nucleotide analogs and immune modulators, do not provide a sterilizing cure emphasizes the need for innovative ways of designing anti-HBV therapies. Extensive research into using gene therapy for the treatment of HBV infection reported promising outcomes in pre-clinical studies. These include RNA interference (RNAi)-based therapies and engineered nucleases such as zinc finger nucleases (ZFNs), transcription activator-like effector nucleases (TALENs), and the RNA-guided clustered regulatory interspaced short palindromic repeats (CRISPR) and CRISPR-associated (Cas) systems [16,17,18,19]. However, finding safe and effective gene delivery vehicles for these nucleic acid sequences remains one of the toughest hurdles in anti-HBV gene therapy. To target CHB, long-term expression of anti-HBV effectors is essential. Whereas non-viral vectors such as lipid-based nano-carrier formulations have several attractive features, they result in transient therapeutic effects [20,21]. The features of rAAVs that have made them one of the leading gene delivery platforms in recent years include rare integration into the host genome, high transduction efficiencies in multiple tissues, including the liver, and sustained transgene expression. The existence of multiple serotypes and artificially synthesized variants gives AAV-derived vectors versatility and several candidates have now been approved for clinical use. 

Glybera was the first AAV-based drug to be approved for clinical application in 2012, which was for the treatment of lipoprotein lipase deficiency. The recent approval of Luxturna, Zolgensma, Hemgenix, Elevidys, and Roctavian for the treatment of retinal dystrophy, spinal muscular atrophy, hemophilia B, Duchenne’s muscular dystrophy and hemophilia A, respectively, further highlight the potential of AAV-based therapeutics. Currently, there are over 130 clinical trials using rAAVs and the feasibility of using AAVs for HBV infection treatment has been demonstrated in pre-clinical studies [16,22,23]. However, high AAV immunogenicity and the resultant toxicity are becoming apparent in non-human primates, as shown in clinical studies [24].

Although AAVs elicit a modest innate immune response when compared to other viral vectors, this is enough to aggravate the adaptive immune response and cause toxicity. A high vector dose is a key factor in most of the studies, which showed AAV-induced toxicities, as demonstrated when using doses higher than 1 × 10^14^ vg/kg to treat neuromuscular disease. The death of four patients who received 1.3 × 10^14^ vg/kg or 3.5 × 10^14^ vg/kg for X-linked myotubular myopathy in a clinical trial (NCT03199469) represents the highest number of deaths reported in a single AAV clinical trial [25]. Although more details are required, the recent deaths of two patients increased the number of deaths in patients who received Zolgensma to three, and further raised safety concerns [26]. In a recent FDA-approved clinical trial for the treatment of Duchenne’s muscular dystrophy (DMD), the administration of a high dose (1 × 10^14^ vg/kg) of rAAV containing a dead *Staphylococcus aureus* Cas 9 (dSaCas9) led to the death of one patient [27]. Several studies have implicated vector-specific immune responses as the major cause of most of these fatalities [24,28,29]. Extensive research has recently focused on understanding AAV immunotoxicity and finding ways to overcome this challenge. Methods adopted include manipulating the host’s immune system or modifying the AAVs to evade immune detection. This review summarizes current knowledge on rAAV immunogenicity and toxicity, discusses progress in overcoming these challenges, and explores how these developments and current knowledge on HBV biology and immunity can benefit endeavors in rAAV-based anti-HBV therapy development.

## 2. Biology of HBV

HBV is genomically diverse, with 10 genotypes (A-J) and various sub-genotypes [30]. The virion genome is located within an icosahedral capsid, which, in turn, is encased in a lipid membrane to constitute a spherical viral particle of 42 nm. The genome is made up of 3.2 kb of relaxed circular DNA (rcDNA) with partially formed positive and complete negative strands. On the 5′ positive strand is a pre-genomic messenger RNA (pg-mRNA) fragment serving a primer function during positive DNA strand repair. The genome carries four overlapping open reading frames (ORFs): *pre-core/Core* (*pre-c/C*), *Pol*, pre-*surface/Surface* (*pre-s/S*), and *X*. These ORFs bear seven translation initiation codons. *Pre-core/C* encodes the hepatitis B core (HBcAg), which forms the viral capsid, along with a secreted hepatitis B e-antigen (HBeAg). *Pol* encodes viral polymerase, while pre-*s/S* makes three envelope structural proteins, referred to by size: large (L-HBs), middle (M-HBs), and small (S-HBs) surface antigens. *X* produces the HBx protein, which is critical for the episomal maintenance of the viral genome. HBV entry and genome transfer into the nucleus of hepatocytes is initiated by binding to the heparan sulfate proteoglycans (HSPGs) at the cell surface. Virion uptake by binding to a receptor, the sodium taurocholate co-transporting peptide (NTCP), in an endocytosis-dependent manner follows. Infected hepatocytes repair the partly double-stranded genome in the nucleus and produce stable covalently closed circular DNA (cccDNA), which serves as a template for viral gene expression. From the cccDNA, pre-genomic RNA (3.5 Kb), viral surface RNAs (2.4 Kb and 2.1 Kb), and *HBx* RNA (0.7 Kb) are produced. These are required for viral replication, capsid formation, episomal maintenance, and infectivity [31]. 

## 3. Immune Response to HBV

The host immune responses not only control the spread of HBV but also contribute to liver inflammation and damage while fighting off the infection. In the early stages of HBV infection, innate immune responses are triggered by viral nucleic acids and viral proteins, leading to the production of antiviral cytokines (e.g., IFN-α) and natural killer (NK) cell activation to impede the initial spread of HBV [32]. The innate immune response is essential for the activation of adaptive immunity. While innate immunity is important for the rapid control of viral replication, the presence of an increase in the number and strength of HBV-specific T (e.g., cytotoxic or helper T cells) and B cells during infection is the ultimate determinant of the outcome of infection. The production of cytokines by CD4^+^ T cells is necessary for the development of CD8^+^ cytotoxic T lymphocytes, which clear HBV infection by destroying the infected hepatocytes [33,34]. 

CHB is often associated with poor effector T-cell responses. This immune ‘exhaustion’ is characterized by poor cytotoxic T-cell activity and impaired cytokine production [35]. The role of regulatory T cells (Tregs) in this ‘immune exhaustion’ has been widely demonstrated [36,37,38,39]. In a poorly understood mechanism, HBV infection induces Treg responses [40]. Recent studies have suggested the involvement of host factors such as heat shock protein 60, furin protein, and viral proteins such as HBeAg and HBx [40,41,42,43]. The study of anti-HBV adaptive immunity has primarily focused on the T-cell responses; however, the various immune functions of B cells, including antigen presentation, antibody secretion, and immune regulation mediate HBV persistence, tolerance, and liver damage [44]. 

## 4. Gene Therapies against HBV

The inhibition of HBV replication by the exogenously supplied mimics of RNAi intermediates, such as artificial primary microRNA (apri-miRNA), short hairpin RNA (shRNA), and short interfering RNAs (siRNA), has been clearly demonstrated. These RNAi activators recruit target HBV RNA to the RNA-induced silencing complex (RISC) for degradation (see Figure 1) [17,20,45,46]. The highly stable replication-competent cccDNA minichromosome persists in infected cells for an extended time and is the major contributor to relapse following treatment withdrawal. Hence, using designer nucleases to inactivate cccDNA permanently has become an attractive option for novel anti-HBV gene therapy development. The engineered nucleases are similar, in the sense that they comprise a DNA binding domain and a nuclease domain. The DNA binding domain in the case of HBV binds to cccDNA or rcDNA, whereas the nuclease domain creates site-specific DNA double-stranded breaks. This recruits repair machinery and, more often, non-homologous end-joining is favored over homology-directed repair, due to the lack of a donor sequence (see Figure 1) [47].

ZFNs typically consist of three to six ZFs that are capable of binding to a DNA sequence of around 9–18 bp [48]. Cleavage of the target DNA is dependent on two ZFN monomers to form an active nuclease (see Figure 1). ZFNs are not commonly used in the field of anti-HBV gene therapy due to the difficulty of constructing them. Two studies have reported on the use of ZFNs against HBV [49,50]. While reduction of viral replication markers was noted, targeted mutagenesis against cccDNA was not present. TALENs differ slightly from ZFNs as they comprise 17.5 tandem repeats comprising 34 amino acids. The amino acids at positions 12 and 13 are referred to as repeat variable diresidues, which are capable of recognizing the signal nucleotides of target DNA. Similar to ZFNs, TALENs function as pairs for site-directed double-stranded DNA cleavage to occur (see Figure 1). Studies using anti-HBV TALENs demonstrated the targeted disruption of HBV DNA [51,52]. To improve specificity, second-generation TALENs make use of mutated FokI nuclease domains as obligate heterodimeric pairs. A study utilizing second-generation anti-HBV TALENs reported the same effectiveness at targeting cccDNA as first-generation TALENs, with improved specificity [53].

CRISPR consists of two components involving single-guide RNA (sgRNA), which forms a complex with the Cas 9 protein to direct cleavage at the target site [54]. The sgRNA is composed of CRISPR-RNA (crRNA) and trans-activating crRNA (tracrRNA) and guides the Cas 9 nuclease to the target site (see Figure 1). The most common Cas 9 nucleases used are taken from *Streptococcus pyogenes* (*S. pyogenes*) and *Staphylococcus aureus* (*S*. *aureus*) [55,56]. CRISPR/Cas 9 has been at the forefront of HBV gene therapy, with numerous studies showcasing its ability to target cccDNA. Several studies have demonstrated significant cccDNA cleavage and reduction of circulating HBV particles in vitro and/or in mouse models [18,57,58,59,60,61,62,63,64]. While off-target cleavage and immunity against the Cas protein is a concern, there is significant progress in developing CRISPR/Cas systems with improved specificity, designing methods to detect off-target effects, and identifying Cas or Cas-like proteins with low seroprevalence [65,66,67,68,69,70]. The progression of EBT-101, a CRISPR/Cas therapeutic against HIV in the Phase I/II clinical trial, highlights the potential of CRISPR/Cas therapeutics against viral infections (https://clinicaltrials.gov/study/NCT05144386, accessed on the 24 November 2023). 

## 5. Biology of AAVs 

AAVs are small, non-enveloped viruses belonging to the Dependoparvovirus genus of the Parvoviridae family of viruses, in which the AAV serotype 2 (AAV2) serves as a prototype. Encapsulated in the ~26 nm capsid is a single-stranded genome of about 4.7 kb, with inverted terminal repeats (ITR) flanking *rep* and *cap* ORFs. Embedded within the *cap* ORF is the assembly activation protein-encoding sequence. The ITRs contain the *cis*-acting Rep protein binding element (RBE), a terminal resolution site (TRS), and three pairs of complementary palindromes (A-A’, B-B’, and C-C’), as well as a single non-palindromic region (D). The palindromes allow for the genome’s T-shaped hairpin secondary structure to form, either in a flip or flop configuration. The ITRs’ *cis*-acting elements and TRS are essential for genome replication, packaging, chromosomal integration, rescue, and circularization, among other functions. A trio of promoters (p40, p5, and p19) drives gene expression from the *rep* and *cap* ORFs. The Rep78, Rep68, Rep52, and Rep40 proteins encoded by *rep* ORF are named according to their molecular size and serve different functions during genome replication and packaging. *Cap* ORF encodes the capsid proteins, VP1, VP2, and VP3 (see Figure 2) [71,72,73].

The high tropic heterogeneity of AAV capsids, with 13 known AAV serotypes and several variants, is mediated by their different primary receptors. Moreover, there is a handful of secondary receptors that co-mediate transduction and infectivity by AAVs [74,75]. The AAVs’ replicative incompetence leads to viral latency post-infection and genome integration at a preferential site, AAVS1, within the host genome [76]. However, AAVs benefit from helper gene expression by adenovirus (Ad) or the herpes simplex virus (HSV), which enables productive infection [77]. 

## 6. AAV-Derived Vectors and Their Application in Anti-HBV Gene Therapy

Besides tropism, the transgene cassette size, duration, and rate of expression need to be considered when designing a viral vector for therapeutic application. The ability of AAVs to form stable, episomal, and circular or concatemeric genomes in the nucleus enables AAV persistence and long-lasting gene expression [78]. All rAAVs have been developed as vectors for gene therapy by deleting both the *rep* and *cap* ORFs, thus maintaining only the ITRs and allowing the insertion of transgenes (see Figure 2). The deleted sequences and helper sequences are supplied in trans during vector production. The single-stranded AAV (ssAAV) and self-complementary AAV (scAAV) constructs used in various studies mimic the genome conformations that occur during the AAV replication cycle. Although the construction of ssAAVs only involves replacing the viral genes, scAAVs also require the mutation of the TRS to abolish nearby ITR resolution [79]. scAAVs thus carry at least three T-shaped hairpin loops (see Figure 2). Compared to scAAVs, ssAAVs have twice the transgene capacity. ssAAVs can accommodate cassettes of about 4.7 kb, but one drawback is their delayed transgene expression compared to scAAVs. This is the result of a requirement for a rate-limiting second-strand synthesis step prior to transgene expression [80]. scAAVs are designed to have a ready-to-express double-stranded conformation, which results in more efficient transduction and faster transgene expression kinetics [81]. 

The scAAV-mediated delivery of anti-HBV shRNA or apri-miRNA has been extensively explored in preclinical studies, with promising outcomes. Targeting the *S* or *X* ORFs using scAAVs expressing shRNA or apri-miRNA in preclinical studies resulted in the inhibition of HBV replication by up to ~98% [22,46,82,83]. As a result of difficulties with designing ZFNs and the large size of TALENs, research on their delivery using AAVs has been limited. One study demonstrated rcDNA-targeted mutagenesis and a reduction in HBV viral particles when using scAAVs expressing anti-HBV ZFNs against HBV *P*, *C*, or *X* genes in cultured cells. However, cytotoxicity was observed with one of the ZFNs [50]. Although the delivery of TALENs targeting HBV has not been reported, a dual-AAV approach with ssAAVs is an option (see Figure 2). 

An all-in-one ssAAV comprising a saCas9 and sgRNA targeting *S* or *P* ORF resulted in the efficient inactivation of HBV replication and the mutation of cccDNA in HepG2-hNTCP cells, a liver-derived HepG2 cell line overexpressing hNTCP that is infectable by HBV [16]. A similar study utilized sgRNAs targeting all HBV ORFs and demonstrated a decrease in HBsAg and HBeAg in vitro [84]. Many studies have now focused on the feasibility of using CRISPR/Cas 9 in vivo and have shown promising results. CRISPR/Cas 9 targeting the *C* and *X* ORF resulted in a significant reduction of HBV DNA in mice [19,85,86] (see Figure 2). Humanized mice can produce cccDNA and present a useful model of HBV infection. The demonstration that the AAV-mediated expression of HBV-targeting sgRNAs can reduce cccDNA in humanized mice further reinforced the suitability of AAVs for anti-HBV gene editor delivery [18].

## 7. Toxicity of rAAV

Features within both the AAV genome and capsids contribute to AAV toxicity. Certain *cis*-regulatory elements found in commonly used non-tissue specific promoters, such as cytomegalovirus (CMV) and chicken beta actin (CAG) promoters, p53 binding elements within the ITRs, and ITR promoter activity have been linked to toxicity [87,88,89,90]. rAAV integration-mediated oncogenicity has been controversial, with contradicting information. However, it is clear that this is a rare event and seems to be dominant in neonatal mice injected with high doses of rAAVs carrying strong promoters [91,92,93]. The majority of these integrations occurred within the microRNA (miR)-341-encoding region within the *Rian* locus of mice, while humans have no ortholog of this miR. Evidence of AAV integration in humans has been of the wild-type AAV integration in the liver. These integrations were also enriched in non-HCC tissues, suggesting that they are not major contributors to HCC in humans [94]. Although genotoxicity has not been demonstrated in long-term AAV-based gene therapy clinical studies, a demonstration that certain enhancer–promoter elements can also facilitate rAAV integration in the human liver will necessitate the careful selection of promoters to drive transgene expression [76]. Whereas genotoxicity and cytotoxicity are a challenge, immunotoxicity appears to be the major obstacle to successful clinical studies and will be the focus of this review. 

## 8. Immune Response to rAAV

The initial, short-lived, and non-antigen-specific innate immune response to AAVs is most commonly initiated by the interaction between the pathogen-associated molecular patterns (PAMPs) of the AAV capsid or the genome (e.g., unmethylated CpG islands in the dsDNA) and the host’s pathogen recognition receptors (PRR), such as the Toll-like receptors (TLRs) on the cell surface (e.g., TLR9, which binds to CpGs, and TLR2, which binds to the AAV antigen). PAMP–PRR interaction results in the activation of transcription factors such as nuclear factor kappa B (NF-κB) and interferon regulatory factors. This mediates the expression of inflammatory cytokines and chemokines, the maturation of dendritic cells into antigen-presenting cells (APCs), and the activation of innate immune cells such as macrophages, NK cells, and neutrophils [28,95]. 

In concert with adaptive immune response factors, complement system activation has been observed in patients who received AAVs [96]. The complement pathway is a component of the innate immune system, which comprises about 50 plasma proteins that interact to activate the inflammatory responses and target pathogens for killing [97]. Although not well understood, AAV binding to antibodies results in the activation of the complement system. This results in the direct killing or neutralization of the AAVs, inflammation, and tissue injury [98,99]. The induction of endoplasmic reticulum stress by AAV transgene products, such as Factor VIII, which is expressed from strong promoters, can also activate an inflammatory response via a poorly understood mechanism [100,101].

The prolonged humoral adaptive immune response to AAVs is initiated by the recognition of the AAV capsid by B cell receptors on naïve or memory B cells, which is followed by antibody secretion. Naïve B cells can also differentiate into short-lived plasma cells that secrete immature antibodies. Alternatively, T helper cells can mediate B cell maturation and differentiation into long-lived plasma cells and memory B cells. Upon antigen re-encounter, these cells secrete high-affinity antibodies, which can then bind to the AAV capsid, resulting in the opsonization and removal of immune complexes by macrophages. B cell responses against capsids do not typically induce tissue damage, but they can neutralize the vector upon re-administration [102]. 

The cellular immune response to AAVs is initiated by T cell activation by peptides from the intracellular degradation of AAV proteins. The peptides bind to MHC-associated molecules before their translocation to the cell surface for presentation to CD8^+^ T cells (MHC I) or CD4^+^ T cells (MHC II). Activated CD4^+^ T cells can differentiate into several Th cell subsets, e.g., Th1 cells differentiate to promote CD8^+^ T cell responses, Th2 cells differentiate to drive B cell activation, and Th17 cells differentiate to activate neutrophils or regulatory T cells to dampen T cell responses. Activated CD8^+^ T cells release cytokines such as IFN-ɣ or lyse APCs by releasing perforin and granzyme B [103,104]. To maintain the balance between self-tolerance and autoimmunity, Tregs suppress the proliferation and/or effector functions of CD4^+^ T cells, CD8^+^ T cells, B cells, NK cells, and antigen-presenting cells (APCs) [105,106,107]. 

## 9. Immune Response to AAVs in the Liver

The liver contains the largest population of NK cells and macrophages, called Kupfer cells (KCs), which directly clear foreign molecules that enter the portal circulation through the gut. However, unlike other macrophages, in most cases, KCs remain in an anti-inflammatory state and are poor activators of T cells. NK cells can serve both immunosuppressive and immune-activating functions through their inhibitory and activating receptors [108,109]. It is well established that liver Tregs mediate immune tolerance via several antigen-specific and non-specific mechanisms, which include the suppression of effector T cells, B cells, and APCs [109]. Programmed death ligand-1 (PD-L1)-expressing and IL-10-producing KCs drive Treg expansion and mediate the conversion of antigen-specific effector T cells into Tregs (see Figure 3) [110,111].

Hepatocytes comprise 80% of the cells in the liver and carry out diverse functions that include protein synthesis, metabolism, and detoxification. Hepatocytes have antigen-presenting capabilities, and they can interact directly with T cells. It is clear that under steady-state conditions, hepatocytes do not express MHC-II. Interestingly, in chronic liver diseases such as hepatitis, hepatocytes express higher levels of MHC-II molecules and induce the expression of Foxp3 in CD4^+^ T cells (see Figure 3) [110,112,113,114]. In non-parenchymal hepatocytes, the sensing of AAV particles by constitutive TLR2 resulted in aggravated innate responses [115]. Other liver-regulatory cells implicated in modulating immune tolerance include type 1 regulatory T cells (Tr1), regulatory B cells, IL-10-producing liver sinusoidal endothelial cells (LSECs), and IL-10-producing dendritic cells [109]. 

The fact that AAVs can exploit the tolerogenic nature of the liver makes them attractive for liver-directed gene transfer [109]. AAV liver gene transfer can result in the deletion of antigen-specific effector T cells and the induction of Treg activation and expansion, leading to the suppression of transgene product specific effector T cell responses and antibody production [116,117]. Although the mechanisms are not well understood, a demonstration that AAV administration to the liver results in IL-10 production by KCs in response to the transgene product suggests that gene expression from AAVs may enhance Treg expansion in an IL-10 and KC-dependent manner (see Figure 3) [110]. 

## 10. Mechanisms and Evidence of rAAV Immunotoxicity

Complement activation has been implicated in several AAV-induced adverse events that occur 6–12 days after infection. Successful treatment of the majority of patients in previous studies with eculizumab, an antibody that inhibits the activation of the complement pathway, confirmed the involvement of this pathway in AAV-induced adverse effects [118,119]. Using a poorly understood mechanism, AAV doses of 5 × 10^13^ vg/kg or higher resulted in complement activation and thrombotic microangiopathies (TMA) in 9 out of 1400 patients in a spinal muscular atrophy (SMA) trial. The majority of those who suffered side effects received a higher dose of 1.1 × 10^14^ vg/kg [28,96,120]. TMA was also observed in 3 patients who received 1–3 × 10^14^ vg/kg and 1 patient who received 5 × 10^13^ vg/kg in two DMD trials that enrolled a total of 15 patients. Although the majority of patients in these trials were treated successfully, one patient in the SMA trial and one in the DMD trial died [24,28,121]. 

The prevalence of AAV-neutralizing antibodies (nAbs) varies, depending on the geographical location and serotype, with anti-AAV serotype 2 (AAV2) nAbs having the highest prevalence worldwide [122]. Cross-reactive antibodies that recognize several serotypes have also been identified. A longitudinal study of patients who received the AAV2 vector at varying doses for the treatment of hemophilia did not highlight serious safety concerns. However, a consistent increase in nAbs to AAV2, as well as AAV5 and AAV8, was noted [123]. Except for the involvement of AAV antibodies in complement activation-associated toxicities, information on the role of capsid-specific nAbs in toxicity is lacking. However, antibody responses to the transgene product have been shown to be toxic and seem to depend on the route of administration. Unlike intramuscular delivery, sub-retinal injection, portal vein administration, hepatic gene transfer, and injection into the central nervous system do not seem to induce transgene-specific antibody production and adverse effects [124,125,126,127].

Most AAV-mediated adverse effects are associated with cellular immunity. AAV capsid antigens are the most common inducers of aggressive adaptive immune responses. Both primary and secondary T cell responses have been implicated in hepatotoxicity following AAV infection. An increase in AAV-specific memory CD8^+^ T cells and diminished transgene expression were detected 4 weeks after infection with Factor IX-expressing AAVs in hemophilia B trials [128,129]. This delayed response suggests that the major contributor to hepatotoxicity is the slow primary T-cell responses driven by the capsid antigens from viral particle degradation. In the SMA trial, one-third of the patients showed an increase in CD8^+^ T cells and liver damage [28,130]. 

## 11. Overcoming AAV Immunotoxicity

### 11.1. Manipulating the Host’s Immune Response

Several studies have demonstrated the potential of general immunosuppressants in rAAV gene therapy [131,132]. A recent study demonstrated that antigen-specific tolerance can be induced by the co-administration of rAAVs with rapamycin. In this study, the co-administration of rapamycin, an inhibitor of the mammalian target of rapamycin (mTOR), with rAAVs resulted in capsid-specific effector T cell depletion and induced Treg proliferation [133]. The use of hydroxychloroquine, a drug used to treat malaria, to inhibit TLR9 activity an hour before AAV administration has also shown promising outcomes [134]. Complement inhibition with agents such as eculizumab and a synthetic peptide APL-9 has been successfully achieved to alleviate AAV-induced toxicity in clinical trials [29,118,119,135]. The administration of IgG-degrading enzymes before AAV injection may also avoid complement activation [136,137]. A recent study generated AAV-specific chimeric antigen receptor (CAR) Tregs. The injection of these CAR Tregs a week after the administration of a highly immunogenic AAV capsid suppressed effector T cell proliferation and cytotoxicity in mice (see Table 1) [138]. Although promising, general immune modulation is less favored in a clinical setting, as it may predispose patients to other infections.

### 11.2. AAV Genome Engineering 

Supported by the superior efficacy of the FIX-Padua variant, highly efficient transgenes can be used to lower the AAV dose [139]. Lower hepatocyte-restricted transgene expression is key to tolerance. Moreover, trials using hepatocyte-specific promoters to express FIX and FVIII from AAVs showed a lack of transgene-specific immune responses and emphasized the need to use weaker liver-specific regulatory elements to drive transgene expression [140,141]. Using *cis*-acting regulatory modules that contain conserved clusters of transcription factor binding sites with high tissue-specific gene expression results in enhanced transgene expression [159]. A recent study reporting superior anti-HBV effects when using a liver-specific promoter to express saCas9 is encouraging (see Table 1) [86].

Recent studies have harnessed RNAi to design self-regulating AAVs and avoid transgene expression in specific cell types. Vectors can be made to carry a target of the tissue-specific miRNA or can be designed to co-express tissue-specific promoter-driven shRNA or pri-miRNA together with transgene mRNAs bearing the target site to the encoded shRNA/pri-miRNA [142]. This concept has been clearly demonstrated in CNS-targeting AAVs. Dorsal root ganglion (DRG) targeting by AAVs is a well-established contributor to AAV toxicity [160,161]. A recent study demonstrated that AAVs bearing dorsal root ganglion (DRG)-specific miR targets in the 3′ untranslated region of the transgene mRNA can mitigate AAV-associated DRG toxicity without affecting transduction in other parts of the brain [162]. A recent study showed that the AAV-mediated delivery of short immunomodulatory peptides, such as the hepatitis C virus non-structural protein 5A (NS5A), suppressed the unwanted activation of primary T-cells ex vivo. The expression of this proline-rich hydrophilic, 20-mer peptide represses the activation of memory T cell responses in the absence of CD3 (see Table 1) [147]. Designing anti-HBV AAVs that co-express this peptide may provide a strategy to overcome the anti-AAV adaptive immune response.

The depletion of CpG motifs in expression cassettes has been shown to diminish innate immune responses and enhance transgene expression [144,145]. The incorporation of TLR9-inhibiting non-coding DNA oligonucleotides into the AAV genome has previously been shown to significantly reduce innate immune and T cell response activation in mice and pigs. Although these results were promising, the vectors could only delay the onset of an inflammatory response in non-human primates, suggesting more complex anti-AAV innate immune responses in this model [163]. The double-stranded RNA molecules produced from the AAV ITR promoter trigger innate immune response-mediated toxicity [90,164,165]. Engineering the expression cassette, such that transcription from the 3′ITR is blocked, is a promising approach to avoid double-stranded RNA formation (see Table 1) [146].

As the only *cis* elements in rAAVs needed for viral replication, ITRs are the primary candidates for vector genome optimization to avoid immune-stimulatory effects. Although ITR modification is complicated by the inflexibility of the ITR sequences and their structure, the mutation of CpG motifs within the ITRs did not affect the therapeutic effects [143]. Interestingly, scAAV vectors induce higher transgene-specific CD8^+^ T and B cell responses than ssAAV [166]. This may be mediated by the double-stranded nature of the scAAV genome, which results in recognition by PRR, the activation of innate immune responses, and/or quicker transgene expression kinetics. 

### 11.3. AAV Capsid Modification

The next generation of recombinant AAV vectors for gene therapy will benefit not only from emerging genome manipulation technologies but also from the improved ability to engineer capsids. Recent high-throughput single-cell analysis studies have unraveled the association between capsid tropism and the upregulation of specific genes, thereby establishing the genetic basis of AAV capsids’ promiscuous cell tropism [167]. Hence, capsid engineering to mediate tissue specificity may reduce the dose needed, de-target APCs, and evade nAbs. Thus far, several studies have demonstrated a range of modification possibilities while retaining capsid integrity, vector assembly, and high production yields. Bioengineering procedures to design capsids with higher hepatocyte transduction efficiencies, including rational designs to modify naturally occurring capsids and the design via directed evolution of novel *cap* genes, have been described and reviewed in the past [135,148,149,150,151]. 

The use of capsid engineering approaches to address the challenge of humoral immune responses and avoid complement activation has been investigated. These include using capsids with modified antibody epitopes and the production of novel capsid variants that can evade pre-existing antibodies via directed evolution [152,153,154]. The development of strategies to circumvent innate and cell-mediated immunity has recently gained momentum. A recent study showed the successful insertion of a TLR signal-inhibiting short peptide in the capsid without interfering with capsid assembly and vector yields. This capsid variant showed reduced Type 1 interferon and CD8^+^ T cell responses and delayed the formation of AAV-specific antibodies [155]. Phosphorylation and the subsequent ubiquitination of tyrosine residues on the AAV capsid enhance capsid degradation and antigen presentation. Hence, the mutation of the surface-exposed tyrosine residue can reduce T-cell activity and the associated toxicity [156,157]. The induction of tolerance by incorporating the MHC II epitopes derived from IgG in the AAV capsid also led to the proliferation of Tregs and the inhibition of CD8^+^ T cell activity (see Table 1) [158]. These techniques have provided limitless opportunities for capsid modification, aimed at overcoming the intrinsic challenges affecting the approval of AAV-based therapies. Although not yet leveraged, these strategies may benefit anti-HBV gene therapy designs. 

## 12. Implications for AAV-Based Anti-HBV Gene Therapy Development

The demonstration that HBV infection enhances AAV transduction in an HBx-dependent manner supports the suitability of using AAVs for anti-HBV gene therapy [168]. AAVs are most likely to be safer and more effective for treatments targeting viruses that accumulate within immunosuppressive or immunotolerant environments, such as the liver. The demonstration that the simultaneous delivery of the same antigen to the liver and muscle can induce hepatic immune tolerance and mitigate the inflammatory responses induced by intramuscular gene transfer is an exciting observation [169,170]. The fact that both AAV and HBV can modulate cellular immunity to favor Treg responses may enhance the suppression of immunogenicity against AAV and boost the efficacy of AAVs in HBV-infected livers. AAV-induced IL-10 overproduction by KCs may suppress CD4^+^ T cell overactivity, as a result of HBV-induced MHC II overexpression by hepatocytes and favor CD4^+^ T cell differentiation into Tregs (see Figure 3). However, the correlation between elevated levels of Tregs with increased HBV loads in patients who are chronically infected with HBV is a concern [38,171]. Hence, harnessing the tolerogenic nature of the liver may be more complicated in patients with CHB. The treatment of patients with immunosuppressants before vector administration comes with the risk of making patients more susceptible to infection. However, inducing antigen-specific tolerance with immunosuppressants such as rapamycin may allow for repeat administrations of low-dose anti-HBV AAVs to achieve a therapeutic efficacy that is similar to that of one-off high-dose administration [133]. 

Lower transgene potency may result in the need for high AAV doses. The availability of highly potent anti-HBV sequences will enable the use of low vector doses, thus diminishing dose-dependent immune response-related adverse effects. This has been demonstrated by the promising outcomes reported in several pre-clinical studies using low-dose AAVs expressing anti-HBV gene editing nucleases or RNAi activators [16,22,85]. The superior anti-HBV effects reported when expressing saCas9 from a liver-specific promoter, the significant progress made in designing tools for the identification of CRISPR/Cas-mediated off-target effects, and limiting immunity against Cas 9 makes anti-HBV CRISPR/Cas therapies more attractive. AAV-mediated hemophilia gene therapy studies have shown that an immune response directed to the therapeutic gene product is linked to increased transgene product-mediated cytotoxic T-cell responses [172]. Evidence of the transgene products’ tolerance in the liver, induced by hepatocyte-restricted transgene expression, further supports the feasibility of AAV-mediated anti-HBV gene therapy [173,174]. Moreover, most of these anti-HBV AAV studies have used weaker liver-specific promoters to drive anti-HBV transgene expression. An example is the mouse transthyretin receptor (MTTR) promoter. 

An efficient anti-HBV gene therapy vector must reach the viral reservoirs and persist in the liver. Hence, the availability of naturally existing AAV serotypes and artificially designed AAVs with high liver tropisms and the ability to form stable episomes is an attractive prospect. The resistance of some AAV serotypes to the humoral responses induced by community-acquired AAV infections may reduce complement activation [175]. AAV-based anti-HBV therapy designs may benefit from currently available capsid modification strategies. Capsid engineering to maximize hepatocyte transduction efficiency and de-target APCs is a promising approach. The insertion of TLR-inhibiting peptides and mutating tyrosine residues on the capsid surface to avoid the activation of TLR 9 signaling and antigen presentation, respectively, is another appealing strategy for anti-HBV gene therapy development. However, a recent demonstration that the AAV capsid assembly is stochastic and results in a highly heterogeneous capsid stoichiometry in one preparation is of concern. This may result in different liver transduction efficiencies between preparations and calls for careful consideration during capsid engineering [176]. 

## 13. Conclusions

Information on AAV immunogenicity has been instrumental in fast-tracking AAV-based gene therapy development. AAV genomes that carry transgene sequences with optimized potency against HBV, expressed from tissue-specific promoters, will reduce the vector doses required for clinical application. Anti-HBV gene therapy may benefit from adapting genome and capsid engineering strategies, developed for improving liver targeting, de-targeting from immune cells, and evading antibody and T-cell responses. However, this field of research has yet to take full advantage of these strategies. Information on how the nature of the immune response in HBV-infected tissues affects these efforts is required to inform the advancement of AAV-mediated anti-HBV gene therapy. Combining vector engineering and pharmacological interventions may be more effective in avoiding possible anti-HBV AAV toxicity. Whereas mouse studies have been instrumental in answering several key questions, they have also been misleading regarding some aspects of AAV immunogenicity. Developing better models of HBV infection is therefore crucial before the clinical translation of anti-HBV AAV therapies.

## Figures and Tables

**Figure 1 microorganisms-11-02985-f001:**
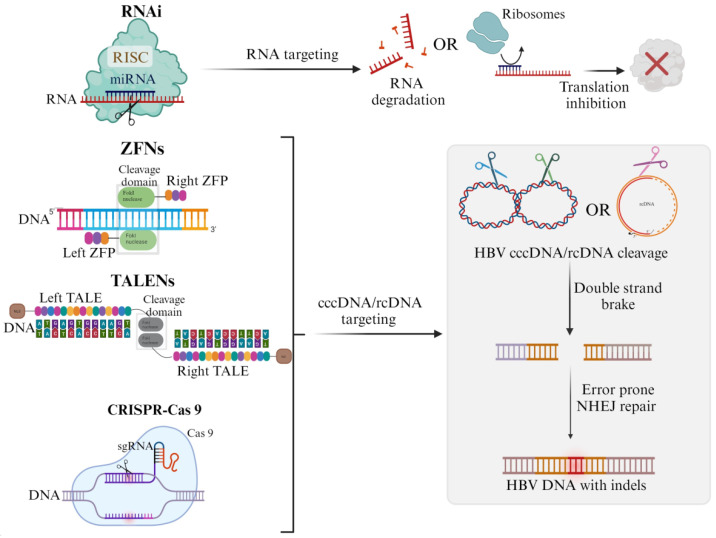
Commonly used tools in HBV gene therapy development. To silence HBV gene expression, the RNAi pathway is activated exogenously by artificial microRNAs (miRNAs), for example. A microRNA duplex is incorporated into an RNA silencing complex (RISC) before the selection of one strand that will guide the RISC to the target HBV messenger RNAs. This results in HBV RNA degradation or translation suppression. Transcription activator-like endonucleases (TALENs) and zinc finger nucleases (ZFNs) work in pairs and require a right TALEN or right ZFN and left TALEN or left ZFN, conjugated to a cleavage domain, for double-strand cleavage to occur. TALENs consist of tandem repeats comprising 33–35 amino acids, whereas ZFNs consist of only 3–6 ZFs. The single guide RNA (sgRNA) associate and directs Cas 9 to the DNA sequence of interest. The Cas 9 enzyme creates double-stranded DNA breaks. Once cleavage has occurred, a double-strand DNA break occurs and recruits host machinery for repair. The error-prone non-homologous end joining (NHEJ) repair pathway is favored, which leads to insertions and deletions (indels) within the cccDNA/rcDNA (created with Biorender.com, accessed on 29 November 2023).

**Figure 2 microorganisms-11-02985-f002:**
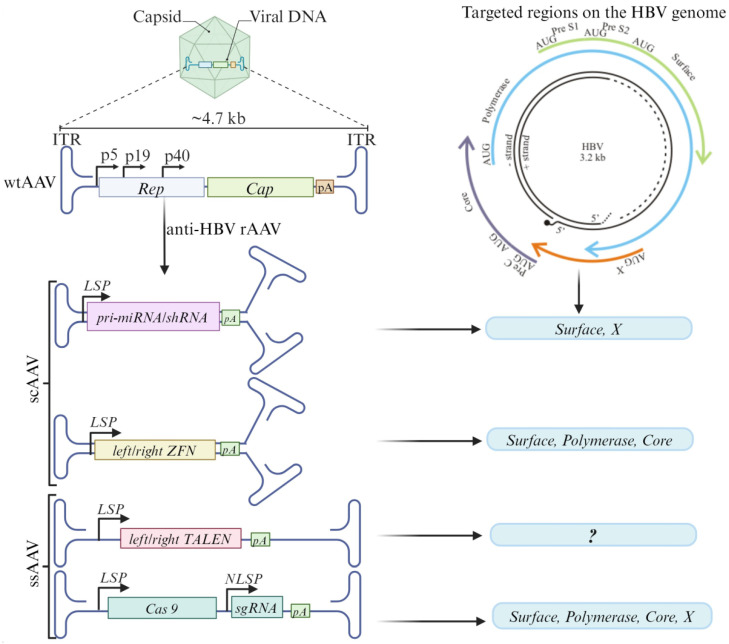
Development of AAVs for anti-HBV gene delivery. The wild-type AAV *rep* and *cap* ORFs are replaced with the anti-HBV sequence, most preferably driven by a liver-specific promoter (LSP). For primary microRNA (pri-miRNA) or short hairpin RNA (shRNA) and zinc finger nucleases (ZFNs), the self-complementary AAVs (scAAVs) are commonly used, whereas single-stranded AAVs (ssAAVs) are used for transcription activator-like endonucleases (TALENs) and RNA-guided clustered regulatory interspaced short palindromic repeats (CRISPR) and CRISPR associated (Cas) sequences. sgRNA expression is commonly driven by a polymerase III non-liver-specific promoter (NLSP). The AAV-mediated delivery of RNAi activators targeting *Surface* and *X* ORFs has been well characterized. AAVs carrying ZNFs targeting *Surface*, *Polymerase*, and *Core* ORFs have been reported. Although dual ssAAVs are promising for TALEN delivery, no anti-HBV TALEN expressing AAV has been reported. AAVs have been used to deliver CRISPR/Cas sequences against all HBV ORFs. The colored arrows on the HBV genome indicate the four open HBV reading frames, with all seven start codons and encoded proteins indicated (created with Biorender.com, accessed on 29 November 2023).

**Figure 3 microorganisms-11-02985-f003:**
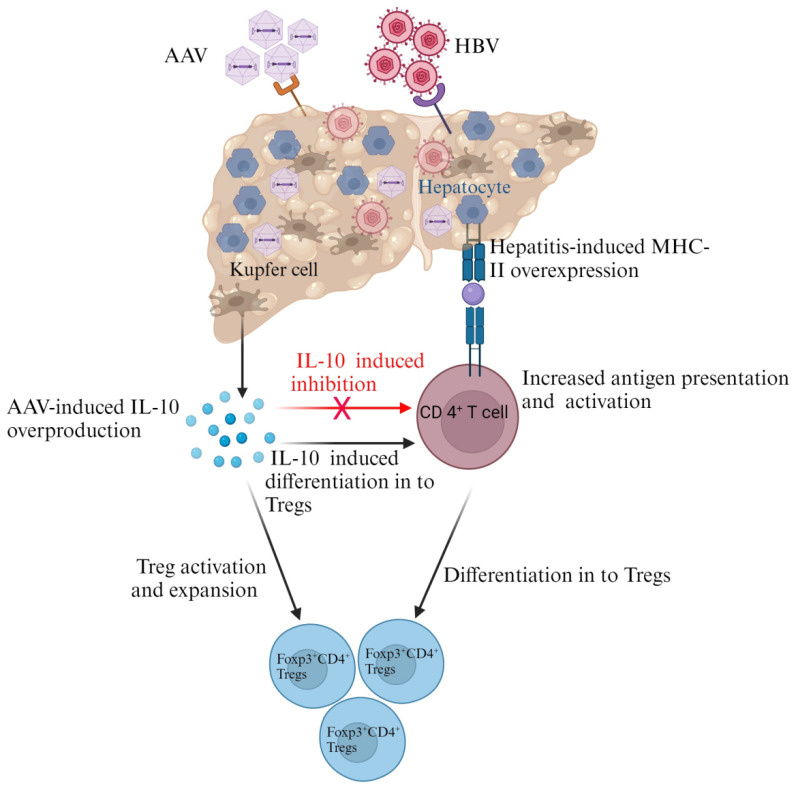
Treg response modulation by AAV and HBV in the liver. Liver infection with AAV increases IL-10 production by KCs and enhances Foxp3^+^CD4^+^ Tregs differentiation and expansion. Secreted IL-10 can also mediate the conversion of antigen-specific CD4^+^ T cells into Foxp3^+^CD4^+^ Tregs and inhibit CD4^+^ T cell activity. Hepatitis, caused by HBV infection, stimulates MHC-II expression by hepatocytes. This enhances antigen presentation to naïve CD4^+^ T cells and results in their activation and differentiation to form Foxp3^+^CD4^+^ Tregs (created with BioRender.com accessed on 21 November 2023).

**Table 1 microorganisms-11-02985-t001:** Approaches to circumventing AAV-mediated immunotoxicity.

Target	Goal	Strategy	References
Host			
	Deplete effecter T cells and induce Treg expansion	Co-administration of AAV with rapamycin	[133]
	Inhibit TLR9 signaling	Hydroxychloroquine administration before AAV injection	[134]
	Inhibit complement activation	Eculizumab or APL-9 or IgG-degrading enzyme administration before AAV injection	[29,118,119,135]
	Induce tolerance	Injecting AAV-specific CAR Treg following AAV injection	[138]
Genome			
	Reduce dose	Efficient transgene engineering	[139]
	Reduce dose and induce tolerance	Transgene expression from weaker hepatocyte-specific promoters and RNAi-mediated transgene self-regulation	[140,141,142]
	Avoid TLR 9 signaling activation	CpG island reduction or TLR9 inhibition via oligonucleotide incorporation in the genome	[143,144,145]
	Avoid dsRNA sensing	Engineering of the expression cassette, such that transcription from 3′ITR is inhibited	[146]
	Repress the activation of primary and memory T-cell responses	Co-expressing NS5A with the transgene	[147]
Capsid			
	Reduce dose, de-target APCs, and induce tolerance	Rational designs and directed evolution to generate capsids with high hepatocyte transduction efficiencies	[148,149,150,151]
	Avoid complement activation	Rational designs and directed evolution to generate capsids that will evade pre-existing antibodies	[135,152,153,154]
	Avoid TLR9 signaling activation	TLR9 inhibiting peptides insertion in the capsid	[155]
	Reduce antigen presentation	Mutation of surface-exposed tyrosine residues	[156,157]
	Enhance Treg proliferation and reduce CD8+ T cell activity	Incorporation of the MHC II epitopes derived from IgG in the AAV capsid	[158]

## Data Availability

Not applicable.

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
