# Peer review of "AAV Immunotoxicity: Implications in Anti-HBV Gene Therapy"

_microorganisms, 2023, doi:10.3390/microorganisms11122985_

Round 1

Reviewer 1 Report

Comments and Suggestions for Authors

The Authors have summarized aspects of AAV use within the context of liver toxicity, specifically immune mediate toxicity. The Authors do provide an appropriate assessment of the landscape. In addition, applications of AAV-mediated HBV vaccines are addressed. However, there are alternative approaches to achieve the goal of HBV vaccination that are not mentioned. Also, there are aspects regarding the quality attributes and challenges of developing AAV with peptide insertions and other protein capsid mosaics that impact drug development.

11.    Sections 8.3-8.5 (lines 349-409). Note that these approaches do appear to be effective for improved AAV performance; however, these approaches may present regulatory and technical challenges to ensure AAV produced in this manner is consistent. As previously shown, even standardized approaches to produce AAV generate a population of AAV capsids with heterogeneous viral particles (Worner et al. 2021. Doi: 10.1038/s41467-021-21935-5). These approaches to insert peptide tags or recognition sites as part of the production process will require more sophisticated analytical methods to assess product quality.

22.  Given the topic of HBV vaccine development in this work and the concept of vector immunotoxicity, it is surprising that the Authors do not mention or discuss the use of LNPs to address this unmet need. A brief discussion on this counterpoint would be informative.

Author Response

Find attached the cover letter with the responses.

Reviewer 2 Report

Comments and Suggestions for Authors

Overall, this is a balanced review that describes potential use of AAVs for delivering anti-HBV therapeutics and, in principle, CRISPR/Cas nucleases. The authors briefly discuss HBV replication (however, cccDNA, the holy grail of HBV persistence, is mentioned only two times in the whole manuscript, while it is the primary target for site-specific nucleases!). Then the authors briefly introduce the readers to the AAV delivery vehicles, their properties, serotypes, the issue of anti-vector immunity and opportunities for manipulating AAV properties. At last, the authors discuss potential application of AAV for delivering CRISPR into the liver from the position of the liver as an immune-privileged organ and opportunities for using low-expression liver-specific promoters. Yet, there are issues that need to be addressed:

cccDNA as the problem of HBV persistence is not properly addressed

There is no data on other nucleases which have been developed many years before CRISPR was adapted for gene editing applications.

Chapter 5. Innate immunity-mediated adverse effects are not discussed from the perspective of the recent NEJM publication describing a patients death after receiving low dose CRISPR/Cas-encoding AAV

Table 1 is inconsistent in general and hard to understand. What “sites” refers to? The purpose is shown to “do something”, and then in the last line – “intravenously delivered AAV”. What does outcome specifically refers to? A positive or negative effects? These all should be stated and explained, as in the current form the Table is hard to understand.

There are many more strategies for altering AAV tropism than shown in Figure 2, such as the use of aptamers, nanobodies, small molecules, hormones etc. Protein capsid modifications is even mentioned by the authors in Chapter 8.5, but is not depicted in the Figure. These should be corrected.

The issue with the chapter of altered AAV tropism is that it basically not related to the topic which the authors discuss. Through the review they keep the balance to mostly focus on the liver and omit other issues related to other organs. However, here, for some reason, strategies for re-directing AAV into other organs are discussed. I would suggest to completely remove this chapter.

Instead, I would suggest to add more details into the chapter describing technologies for avoiding immune-mediated toxicities and their efficacy, and perspectives

Chapter 9 should be discussed from the position of recent NEJM data describing deadly immune responses to low-dose AAV and to the necessity of introducing constitutive CRISPR/Cas-producing vectors for targeting HBV from the perspective of off-target activity and anti-Cas immunity, in general.

The information about the potential use of CRISPR/cas against HBV is also scarse. The field has been developing for almost 10 years, and there is a remarkable progress in the use of CRISPR/Cas for depleting HBV. I would suggest to add more background information that concerns the use of CRISPR against HBV.

Comments on the Quality of English Language

Minor typos and English check is necessary

Author Response

Attached is the cover letter with the responses.

Reviewer 3 Report

Comments and Suggestions for Authors

In the review “Innate immune Response to viral vectors in Gene Therapy”, Y. Wang and W. Shao focus on the implication of AAV immunotoxicity on anti-HBV gene therapy. The manuscript is well-written and documented. Although there are already many reviews on AAV immunogenicity and how to circumvent them (e.g Ertl Front Immunol 2022; Wang et al. Viruses 2023; Costa-Verdara et al. Hum Gene Ther 2023), this manuscript could still be original and give further insights to both AAV gene therapy and anti-HBV immunity fields. However, to achieve this aim, the authors should address the following main points:

11 -     The title implies a focus on anti-HBV gene therapy but this part is not developed in the manuscript. It would be interesting to have a statement on anti-HBV gene therapy with a table summarizing the current gene therapy approaches for anti-HBV therapies and vaccines and how AAV vectors are relevant. 

22-     The manuscript is mainly focused on AAV biology, immunogenicity/immunotoxicity and strategies to circumvent it. In each part should be added a statement on how each limit or observation related to AAV vector could benefit for anti-HBV gene therapy or be a limit.

33-     The authors give an overview of AAV-related toxicity that happens in patients treated for inherited disorders with high doses of vectors. An anti-HBV gene therapy will probably not require comparable doses and/or serotypes. This point should be developed further.

  4-      A section describing applications of AAV vectors as vaccines might be relevant.

Minor points:

1-     A balance between HBV and AAV should be taken into consideration in regards of the title. AAV is predominant in the current manuscript.

2-     The biology of AAVs section should be shortened.

3-     The review requires more illustrations at least on HBV and AAV structure/biology and Tables to summarize all the information.

Author Response

(The authors gave the same response as above.)

Round 2

Reviewer 2 Report

Comments and Suggestions for Authors

The authors addressed the issues

Reviewer 3 Report

Comments and Suggestions for Authors

The authors addressed all the comments.